# Survival of Breast Cancer by Stage, Grade and Molecular Groups in Mallorca, Spain

**DOI:** 10.3390/jcm11195708

**Published:** 2022-09-27

**Authors:** Maria Clara Pascual, Juan José Montaño, Paula Franch, Carmen Sánchez-Contador, Maria Ramos

**Affiliations:** 1Department of Psychology, University of Balearic Islands (UIB), 07122 Palma, Spain; 2Health Research Institute of the Balearic Islands (IdISBa), 07120 Palma, Spain; 3Balearic Islands Public Health Department, Government of the Balearic Islands, 07010 Palma, Spain

**Keywords:** breast cancer, survival, multiple imputation, competing-risks regression model

## Abstract

The aims of this study are: (1) to determine cause-specific survival by stage, grade, and molecular groups of breast cancer, (2) to identify factors which explain and predict the likelihood of survival and the risk of dying from this cancer; and (3) to find out the distribution of breast cancer cases by stage, grade, and molecular groups in females diagnosed in the period 2006–2012 in Mallorca (Spain). We collected data regarding age, date and diagnostic method, histology, laterality, sublocation, pathological or clinical tumor size (T), pathological or clinical regional lymph nodes (N), metastasis (M) and stage, histologic grade, estrogen and progesterone receptors status, HER-2 expression, Ki67 level, molecular classification, date of last follow-up or date of death, and cause of death. We identified 2869 cases. Cause-specific survival for the entire sample was 96% 1 year after diagnosis, 91% at 3 years and 87% at 5 years. Relative survival was 96.9% 1 year after diagnosis, 92.6% at 3 years and 88.5% at 5 years. The competing-risks regression model determined that patients over 65 years of age and patients with triple negative cancer have worse prognoses, and as stages progress, the prognosis for breast cancer worsens, especially from stage III.

## 1. Introduction

Breast cancer is the most common cancer and the leading cause of death from cancer in women around the world [1], including in Spain [2]. According to REDECAN, the estimated incidence cases in women in 2022 will be 34,750 new diagnoses in Spain [2].

According to EUROCARE-5, for the period 2000–2007, 5-year relative survival with breast cancer in Spain was 82.8% (81.9–83.6), slightly higher than the average in Europe, that was 81.8% (81.6–82.0), with a range from 74% for Eastern Europe to 85% for Northern Europe [3]. Compared with the period 1995–1999, breast cancer survival has improved in Spain [4].

Analyses of prognostic factors on survival are essential for patients and professionals and impact on health policy. Scientific evidence has demonstrated the prognostic value of different variables, such as age, ethnicity, tumor size, histology, histological grade, stage at diagnosis, hormone receptor status, and the surgical and adjuvant treatment patients receive [5,6].

Although stage, grade, and molecular groups are prognostic factors in breast cancer [6], information about survival based on these variables continues to be scarce. Regarding stage, grouped classification (localized, regional extension and metastasized) [7] or four-stage classification (I, II, III and IV) [8,9] are widely used. However, the International Union Against Cancer TNM system (7th edition) classifies invasive breast cancer stage into eight categories (IA, IB, IIA, IIB, IIIA, IIIB, IIIC and IV) [10]. Information about survival by stage, grade, or molecular group, as well as the relationships among these variables, is vital for making clinical decisions. Progressive improvements in survival associated with early detection and better management and treatment of the disease have been observed [6].

The aims of this study are: 1. to determine cause-specific survival by stage, grade, and molecular group of breast cancer; 2. to identify factors which explain and predict the likelihood of survival and the risk of dying from this cancer; and 3. to determine the distribution of breast cancer cases by stage, grade, and molecular group.

## 2. Materials and Methods

### 2.1. Patient Involvement

Our research comprised a population-based, retrospective follow-up study of female patients living in Mallorca, diagnosed with invasive breast cancer (C50) between 2006 and 2012, identified through the Mallorca Cancer Registry. The total population of Mallorca in 2012 was 876,147 inhabitants. We excluded cases exclusively identified through the death certificate (DCO) and cases without follow-up (only for survival analysis).

### 2.2. Variables

The following data were collected: age at diagnosis, date and diagnostic method, histology, laterality, sublocation, pathological or clinical tumor size (T), pathological or clinical regional lymph node status (N), metastasis (M) and stage, histologic grade, estrogen and progesterone receptors status, HER-2 expression, Ki67 levels, molecular classification, date of last follow-up or date of death and cause of death (breast cancer or other causes).

We grouped ages as 15–44 years or age, 45–54, 55–64, 65–74, or 75 and over. Diagnostic method was recorded as clinical or pathological. Histology was grouped as ductal/NST (8010, 8020, 8140, 8500, 8501, 8521, 8541), lobular (8520), other carcinomas/special sub-type (8071, 8200, 8201, 8211, 8246, 8260, 8401, 8480, 8490, 8503, 8504, 8507, 8510, 8513, 8530, 8560, 8572, 8575), mixed carcinomas (8522–8524), and other neoplasms (non-epithelial and non-specific) (8805, 8890, 8980, 8894, 9020, 9180, 8000, 8001). Laterality was recorded as left, right, or bilateral. We classified sublocation as nipple/central region, intern upper quadrant, intern lower quadrant, extern upper quadrant, extern lower quadrant, axillary, or more than one location. The stage was calculated according to the UICC 7th edition, and grouped in the following categories: IA, IB, IIA, IIB, IIIA, IIIB, IIIC and IV. Histologic grade was recorded as well differentiated, moderately differentiated, or poorly differentiated. Estrogen and progesterone receptors (ER and PR) and HER-2 expression were recorded as positive or negative. Ki67 level was recorded as low or high (cut-off: 20%). Finally, molecular classification was grouped as follows: luminal A (ER or PR +, Ki67 low, and HER-2−), luminal B (ER or PR+, Ki67 high, or HER-2+), luminal with ki67 unknown (ER or PR+, Ki67 missing, and HER-2− or missing), HER-2 enriched (ER and PR−, HER-2 +) and triple negative (ER and PR−, HER-2−).

We defined survival time from the date of diagnosis to the date of last known vital status (death by any cause, date of loss to follow-up, or date of the end of follow-up on 31 December 2018). Vital status was categorized as alive (0), dead by breast cancer (1) or dead by other causes (2).

### 2.3. Statistical Analysis

We used the multiple imputation (MI) method to assign values when these were missing in the following main variables: laterality, sublocation, stage, histologic grade, and molecular classification. Three main steps were followed [11]. First, we ran the imputation model and replaced each missing value with sets of 5, 10, 15, and 20 imputations by applying the multiple imputation chained equation (MICE) procedures. We made the MI using sex, age, diagnostic method, histology, time, and vital status. A more detailed description can be found in a previous manuscript [12]. Secondly, we independently analyzed the resulting imputed and complete data sets by applying a competing-risks regression. Finally, we applied a single competing-risks regression model using Rubin’s rules [13] from each set of 5, 10, 15, and 20 estimates resulting from the previous competing-risks regression model. We selected the MI with five imputations because the increase to 10, 15, or 20 did not change the coefficient values, the standard errors, or the degrees of significance.

Before performing the survival analysis, we explored the relationships among the variables using contingency tables, on the basis of which the Chi-square independence test and the Cramer’s V association index were performed.

As we knew the cause of death, we used Cause-Specific Survival (CSS), but we also calculated the Relative Survival (RS) by the Ederer II method [14] using life tables obtained from published official mortality data for the Balearic Islands [15]. Since survival studies such as EUROCARE and CONCORD have used RS, we decided to calculate both survival types in order to be able to compare them with each other and with the aforementioned studies.

We applied the actuarial and Kaplan-Meier methods in our survival analysis to estimate the likelihood of survival and risk of death. We used the log-rank test to evaluate the statistical differences of the observed survival curves by each categorical variable and created graphic representations thereof in order to compare and observe the evolution of survival over time. Finally, we applied competing-risks regression models to identify the prognostic factors associated with mortality risk. The regression model included age, diagnostic method, sublocation, histology, stage, laterality, molecular classification, and histologic grade. Cases at stage IB were excluded because their survival rate was 100%. We tested the proportional hazard assumption for each covariate by introducing time-dependent variables.

Competing-risks regression [16] provides a valuable alternative to Cox regression [17] for survival data in the presence of competing risks. Competing-risks regression posits a model for the subhazards function of a failure event of primary interest in the presence of competing failure events that impede the event of interest. This must not be confused with the usual right-censoring found in survival data, such as censoring due to loss to follow-up. However, while censoring merely obstructs from observing the event of interest, a competing event prevents the event of interest from occurring altogether. In our study, the event of interest was breast cancer death, while the competing failure event was death from other causes. Finally, this model estimates the subhazard ratios in a manner akin to the hazard ratios in the Cox regression.

We selected the covariates in the final competing-risks model using the Wald test. We performed the competing-risks regression model both before and after MI to compare the effect of the imputation procedure on the subhazard ratio estimation of covariates.

We used STATA 16 for MI and CSS analysis and the ‘relsurv’ R library for RS.

## 3. Results

We identified a total of 2885 breast cancer cases with diagnoses between 2006 and 2012. We excluded 16 DCO cases, so the final sample was 2869 cases. Of them, 98.8% were diagnosed by pathological methods (1.2% by clinical methods), and 82.0% had ductal/NST histology. There were 5.4% of cases with unknown laterality, 18.2% with unknown sublocation, 16.7% with unknown T, 17.8% with unknown N, 14.8% with unknown M, 16.7% with unknown stage, 22.7% with unknown histologic grade, and 22.0% with unknown molecular classification. After MI, 30.1% were in stage IA, 3.4% were in stage IB, 24.1% were in stage IIA, 15.7% were in stage IIB, 11.2% were in stage IIIA, 3.8% were in stage IIIB, 3.4% were in stage IIIC, and 8.3% were in stage IV. Table 1 presents a complete description of the sample and the distribution of the variables imputed after applying MI.

A survival analysis was performed with 2867 cases, because 2 cases had no follow-up. At the end of the study, a total of 2042 (71.2%) patients had survived; 522 (18.2%) died of breast cancer, 302 (10.5%) died from other causes, and 3 (0.1%) died from unknown causes. The average survival time was 11.06 years (CI 95% = [10.91, 11.21]), with a standard error of 0.077. CSS for the entire sample was 96% one year after diagnosis, 91% at three years and 87% at five years. RS was 96.9% one year after diagnosis, 92.6% at three years, and 88.5% at five years.

Table 2 shows CSS by stage and year before and after MI. If MI had not been not applied, there would have been a slight overestimation in the initial stages; on the other hand, there would have been an underestimation in the more advanced stages (IIIB, IIIC and IV). Survival times with breast cancer seemed to stabilize for some stages (IB, IIA, IIIB, IIIC and IV) but not for others (IA, IIB, IIIA).

In the same way, Table 3 shows CSS by molecular classification and the year before and after MI. Again, if MI had not been applied, there would have been a slight overestimation in all molecular groups, except in the case of the triple negative, in which there would have been a slight underestimation.

Table 4 shows CSS and RS at 5 years by stage and molecular classification before and after MI. Generally, a slightly higher RS can be observed compared to CSS before and after MI. Conversely, both the SR and the CSS are slightly lower after MI.

Survival curves showed differences in breast cancer survival (*p* < 0.001) by age and histology (Figure 1), stage, histologic grade, and molecular classification (Figure 2). Laterality was significant at *p* = 0.09 and sublocation was significant at *p* = 0.07. Comparing each variable by pair of categories, all age groups presented differences (*p* < 0.05), except between the 15–44 and 45–54 groups and between the 45–54 and 55–64 groups. Breast cancer survival diminishes markedly in people over 75 years of age. Ductal/NST and mixed carcinoma histologies have better survival compared to other carcinoma subtypes (*p* < 0.05); other neo-plasms (non-epithelial and non-specific) have the worst survival. There were survival differences in all stages (*p* < 0.05), except between IA and IB and IIA and IIB. All categories of histologic grade presented differences (*p* < 0.05); the prognosis worsened as the grade of differentiation decreased. All categories of molecular classification presented differences (*p* < 0.05), except between luminal B and luminal with ki67 unknown, and between HER-2 enriched and triple negative, being luminal A the category with the best survival. Slight changes could be seen in the survival curves after applying MI (Figure 2).

The Wald test included age, sublocation, stage, laterality, and molecular classification in the final competing-risks regression model. Therefore, we excluded diagnostic method, histology, and histologic grade. The exclusion of histologic grade was probably due to its relationship with molecular classification (χ^2^(8) = 594.44, *p* < 0.001; Cramer’s V index = 0.38, *p* < 0.001). Table 5 shows the results of the competing-risks model before and after MI. After MI, the model determined that patients over 65 years old had worse prognoses than the 55–64 years old group. Also, patients with triple negative had a worse prognosis than those with luminal with ki67 unknown. As stages progress, the prognosis for breast cancer worsens, especially for stage IV. In general, standard errors were lower after MI, providing more accurate estimates of the risk of dying from this cancer. Finally, sublocation and laterality were no longer significant after MI.

## 4. Discussion

The CSS of breast cancer at 1, 3, and 5 years was 96%, 91%, and 87% respectively, while the RS 1, 3, and 5 years after diagnosis was 96.9%, 92.6%, and 88.5%, respectively. The RS was slightly higher than CSS, as previously observed in other studies for breast and other cancers, such as prostate cancer, for which early diagnosis is performed [18,19]. In these cancers, RS overestimates survival because of earlier diagnosis. In the study by De Lacerda et al., for example, the difference between CSS and RS 5 years after diagnosis (period 2000–2013, N = 653,181 cases) was like ours.

Comparing the RS obtained in our study with those of other population-based studies, we see that the CONCORD-3 (period 2010–2014) obtained a net survival at five years of between 70% and 85%. Most European countries, including Spain, the United States, Canada, and Australia had a survival rate of 85% or more, up to a maximum of 92.8%. In Spain, the CONCORD-3 study concluded that 5-year survival from diagnosis for the period 2000–2004 was 82.9%, while for the period 2005–2009, it was 84.6%, and for the period 2010–2014, it was 85.2% [20].

On the other hand, the EUROCARE-5 study (1999–2007) obtained a European-wide RS 5 years after diagnosis of 81.8% and 82.8% in Spain [3]. In the previous period (EUROCARE-4, 1995–1999), the RS was 79.4% in Europe and 80.3% in Spain [21]. According to REDECAN, survival rates by region showed slight differences: Catalonia obtained a survival rate of 86.2% in Girona in 2005–2009 and 87.1% in Tarragona in the same period. The Basque Country recorded survival of 84.6% in 2000–2012, and the Canary Islands had a survival of 86.6% in 2008–2012. The survival rate obtained in our study is among the highest published to date. The management of breast cancer in Spain is partly associated with the implementation of Population Screening Programs in the 1990s. In our Autonomous Community, the program for the early detection of breast cancer began in 1998, one of the last in the country. Despite this, the data recorded by the Carlos III Health Institute [22] indicate that mortality in our Autonomous Community is among the lowest in the country. We believe that the private sector, which predominates in our region, is compensating for the late start and low coverage of our screening program. According to Grande et al., monitoring of regional epidemiological indicators for breast cancer is crucial to evaluate the different measures taken for breast cancer control [23].

In the CONCORD and EUROCARE studies, survival was shown to be increasing over time. The obtained results, i.e., 88.5% five years after diagnosis, confirmed this trend, which was attributed to the implementation of screening programs and therapeutic improvements [3].

We have observed that the survival of breast cancer cases does not stabilize after ten years; this is consistent with recent cure fraction studies that have shown that after ten years, only 50% of breast cancers are cured and that the time to cure depends on age, being lower in middle age women, and higher in young (15–44 years old) or oldest (65–74 years old) women in some stages but not in others [24]. In our study, survival stabilization could be also related to stage, as we have seen that it occurs after ten years in some stages but not in others.

In our study, breast cancer survival was associated only with age over 65, triple negative, and stage. Furthermore, the competing risks regression model showed that, in contrast, neither sublocation nor laterality affected survival after MI. However, other studies have found a relationship between sublocation and survival [25].

On the other hand, age is a known independent prognostic factor in numerous studies [3]. It has been previously described that the age with the best overall survival is around 50; from there, it goes down with increasing age [26]. In our study, the age range with the best survival was 55–64 years, more or less coinciding with the age range of the population-based breast cancer screening program in Mallorca until 2006. From then on, the age included in the program was progressively extended, reaching 69 in 2011. The age extension has not so far resulted in improved survival in the women included in our study. This situation could be due to a delay in achieving adequate population coverage, due to the limited resources of the screening program. The regression model has shown that being older than 64 affects survival; this is especially the case for women above 75 years of age. Some authors relate these results to suboptimal treatment in this age group due to the presence of comorbidities, possible toxicities, preferences of the patient, etc. [27]. For the purpose of understanding the impact of the screening program (age 50–69 years) on survival, the Appendix A provides the results grouped by age as follows: (1) up to 49 years of age; (2) from 50 to 69 years of age; (3) over 70 years of age (Appendix A). It can be observed that patients who are in the screening program have a better survival rate compared to the group of older women and a similar survival rate to the group of younger women.

Diagnosis stage is a factor that shows the most evident relationship with survival. All stages are significant concerning the reference group (IA), and the prognosis changes as the diagnosis stage progresses. For example, diagnosis at stage II (IIA and IIB) implies approximately double the likelihood of not surviving compared to stage IA. In stages IIIA and IIIB, this probability is multiplied by approximately five times, while the probability of not surviving when with stage IIIC (characterized by positive supraclavicular lymph nodes) is already almost eight times that of stage IA. The worst prognosis is at stage IV, i.e., almost 22 times more likely not to survive than the reference stage. Fortunately, only 26.7% of cases are in stage III or IV. These results demonstrate that the stage at the time of diagnosis is key to patient survival. Moreover, it is a variable on which we can take action, as it is not an intrinsic characteristic of cancer itself. Therefore, it is necessary to develop and maintain early detection programs and rapid diagnosis protocols that allow diagnoses to be made at the earliest possible stage, as this will have a clear impact on patient survival.

To our knowledge, no survival study based on population data according to stage—categorized into eight levels—has been published. However, at the European level, Nordenskjöld et al. presented survival data by stage in a population-based study of diagnoses made between 1989 and 2013 in Sweden (N = 42,220). Those authors observed 5-year survival rates of 97.8% in stage I, 87.4% in stage II with negative nodes, 89.5% in stage II with positive nodes, 64.1% in stage III and 17.1% in stage IV [28]. Nationally, diagnoses between 2000 and 2012 from the Granada Registry had RS five years after diagnosis of 96.6% for stage I, 88.2% for stage II, 62.5% for stage III, and 23.3% for stage IV [29]. In all cases, the survival rates obtained in our sample were slightly higher, especially in stage IV.

Finally, our analysis of the molecular group using the competitive risk model showed that only belonging to the triple-negative group means a worse survival rate than the reference group (Luminal with unknown Ki67), i.e., the risk of dying from breast cancer in this group is almost double that of the other groups. If we analyze CSS and RS by molecular group, the triple-negative and the HER-2 enriched groups had the worst survival. This survival distribution was previously known and has been analyzed in recent years in several population-based studies [30,31]. The best survival rate was observed in the Luminal A group; this was probably due to the fact that this group has therapeutic targets (hormone therapy), while research is still being carried out to determine which therapeutic targets may be effective for the treatment of triple-negative breast cancer. Additionally, triple negative breast cancer is particularly heterogeneous [31].

Regarding the other factors included in our study that may affect survival, although not included as significant in the competing risks model, we obtained histology and histological grade data. Concerning these factors, it is necessary to note that histology likely affects survival, because it is a characteristic of the tumor itself. Hence, the histological group with the worst survival is “other neoplasms” (non-epithelial, non-specific), which includes sarcomas and non-specific histologies, both of which have poor prognoses. On the other hand, histological grade has been shown to be a prognostic factor associated with breast cancer survival, but due to its close relationship with molecular group, we were not able to include it in the competitive risk model [32].

The percentage of missing cases in the stage was 16,8%, apparently higher than that found in the high-resolution CONCORD study, where it was 8% in European registries and 11% in registries in the United States. In some studies, it is assumed that if T and N are known, M can be considered 0 [33]. In our case, in cases where M was unknown at initial diagnosis, a thorough review of the clinical history was performed to confirm, whenever possible, the value of M. Moreover, they decided to exclude unstaged cases. We have shown that multiple imputation of missing stage or molecular groups avoids underestimating survival in advanced stages or triple-negative cases while causing these results to be overestimated in early stages or in other molecular groups. Therefore, the use of multiple imputation made it possible to use data for all of the patients in the database and obtain unbiased and more accurate estimates of breast cancer survival. In this line, Derks et al. identified the need to apply competing-risks models in long follow-up survival studies on breast cancer in which other causes of death are taken into account, as well as its usefulness when we values are missing [34].

One limitation in our study was the relatively high percentage of missing values regarding molecular group, specifically about Ki67, because clinicians did not use this metric during the study period. However, we overcame this by creating the category Luminal with Ki67 unknown to maximize the information available about ER, PR and HER-2. This strategy allowed us to reduce the missing values in molecular groups from 56.8% to 22.0%. Ki67 is an independent prognostic variable which is currently being used in clinical practice to make therapeutic decisions [35]. However, its use is controversial, because there is no single helpful cutoff point [36], and other studies have failed to include in their molecular classifications [37].

Another limitation was that we did not collect information about treatment, relapses, or risk mutations, such as BRCA1-2. It is essential to note the difficulty in collecting this type of information from cancer registries due to the complexity of searching for it in patients’ medical records. We initially tried to collect the treatments received by patients with a sample taken from our database, but due to the complexity involved, we decided to focus on other aspects. In light of this fact, it should be noted that information about TNM is also challenging to find, and, in Spain, the cancer register of Mallorca is one of the few that collect it.

The strength of this study is that the sample was population-based, with high-quality data, as the negligible percentage of missing cases in terms of survival information and the minimal differences observed between cancer-specific and relative survival show. In our research, clinical records of each case were reviewed by trained professionals. We followed some of the cases for up to 13 years. Moreover, we knew the cause of death, which allowed us to calculate cause-specific survival. The application of the competing-risks model instead of the Cox model used in other studies [12] made it possible to obtain more realistic estimates of the risk of dying from breast cancer, considering that there are obviously patients who die from other causes. On the other hand, it was possible to collect a pathological diagnosis in many cases, which is unusual compared to other tumors which are frequently diagnosed by imaging or other clinical tests.

## 5. Conclusions

The breast cancer CSS and RS obtained in this study are good, i.e., above the average in Spain, confirming the improving trends for this cancer. We conclude that age, stage at diagnosis, and molecular classification are significant prognostic factors. Our data indicate that triple-negative tumors have the worst prognosis regarding molecular classification. In addition, our study showed a worsening breast survival rate by stage III, although these cases represent only one quarter of cases. Therefore, reinforcing early detection breast cancer programs and developing rapid diagnosis protocols are essential.

## Figures and Tables

**Figure 1 jcm-11-05708-f001:**
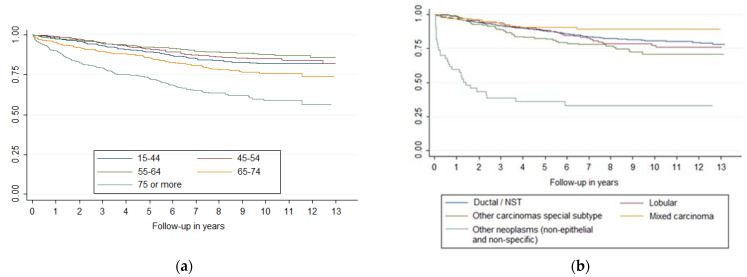
Survival curves of breast cancer by (**a**) age and (**b**) histology.

**Figure 2 jcm-11-05708-f002:**
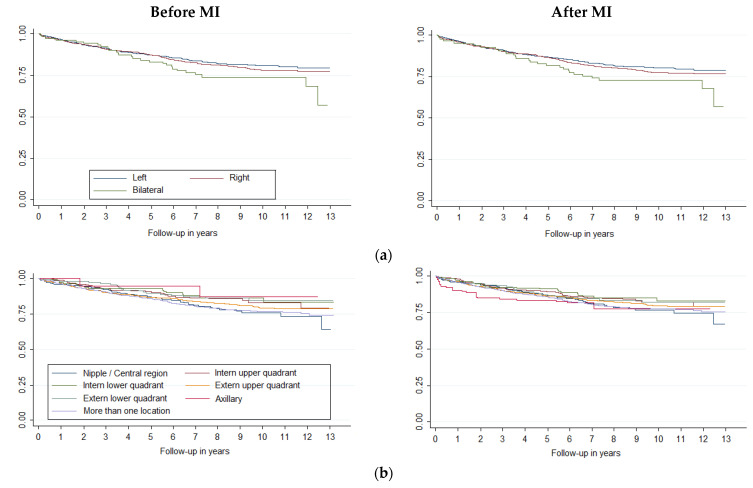
Survival curves of breast cancer by (**a**) laterality, (**b**) sublocation, (**c**) stage, (**d**) histologic grade and (**e**) molecular classification, before and after Multiple Imputation (MI) (m = 5).

**Table 1 jcm-11-05708-t001:** Clinical description of breast cancer cases diagnosed in Mallorca between 2006–2012 (N = 2869).

Variable	Categories	Number	%	% after MI
Age	15–44	478	16.7	
	45–54	687	23.9	
	55–64	673	23.5	
	65–74	454	15.8	
	75 or more	575	20.0	
	Missing	2	0.1	
Histology	Ductal/NST	2353	82.0	
	Lobular	214	7.5	
	Other carcinomas special subtype	151	5.3	
	Mixed carcinomas	99	3.5	
	Other neoplasms (non-epithelial and non-specific)	52	1.8	
Laterality	Left	1325	46.2	49.0
	Right	1287	44.9	47.0
	Bilateral	102	3.6	4.0
	Missing	155	5.4	
Sublocation	Nipple/Central region	221	7.4	9.3
	Intern upper quadrant	206	7.2	8.8
	Intern lower quadrant	115	4.0	4.9
	Extern upper quadrant	772	26.9	32.4
	Extern lower quadrant	144	5.0	6.2
	Axillary	18	0.6	1.1
	More than one location	882	30.7	37.3
	Missing	521	18.2	
Stage	IA	724	25.2	30.1
	IB	78	2.7	3.4
	IIA	587	20.5	24.1
	IIB	385	13.4	15.7
	IIIA	256	8.9	11.2
	IIIB	88	3.1	3.8
	IIIC	75	2.6	3.4
	IV	196	6.8	8.3
	Missing	480	16.7	
Histologic grade	Well differentiated	518	18.1	24.1
	Moderately differentiated	971	33.8	43.7
	Poorly differentiated	730	25.4	32.2
	Missing	650	22.7	
Estrogen receptors	Positive	1798	62.7	
	Negative	465	16.2	
	Missing	606	21.1	
Progesterone receptors	Positive	1441	50.2	
	Negative	813	28.3	
	Missing	615	21.4	
HER-2	Positive	356	12.4	
	Negative	1816	63.3	
	Missing	697	24.3	
Ki67	Low	391	13.6	
	High	485	16.9	
	Missing	1993	69.5	
Molecular classification	Luminal A	349	12.2	15.6
	Luminal B	482	16.8	21.5
	Luminal with Ki67 unknown	999	34.8	44.4
	HER-2 enriched	130	4.5	6.2
	Triple negative	277	9.7	12.3
	Missing	632	22.0	
Vital status at the end of follow-up	Alive	2042	71.2	
	Death from breast cancer	522	18.2	
	Death from other causes	302	10.5	
	Cause of death is unknown	3	0.1	

**Table 2 jcm-11-05708-t002:** Cause-specific survival (CSS) function in percentages by years of follow-up and stage based on the actuarial method, before and after multiple imputation (MI) (m = 5).

	Original Data Set N = 2387									Imputed Data Set N = 2867								
Year	IA	IB	IIA	IIB	IIIA	IIIB	IIIC	IV	Total	IA	IB	IIA	IIB	IIIA	IIIB	IIIC	IV	Total
1	100	100	100	99	98	92	95	69	96	99	99	99	98	96	91	93	73	96
2	99	100	99	98	94	79	83	56	94	99	99	98	97	91	80	81	61	93
3	99	100	98	96	88	76	73	44	91	98	98	97	95	87	77	73	51	91
4	99	100	96	94	85	73	72	36	89	98	98	95	93	83	75	71	43	88
5	98	100	94	93	80	71	63	31	87	97	98	93	92	79	73	64	39	87
6	98	100	92	89	76	67	59	24	84	96	98	91	89	76	69	60	33	84
7	97	100	90	87	72	64	54	21	82	95	98	89	87	72	66	56	29	82
8	96	100	89	86	69	62	52	16	81	95	98	88	86	70	64	55	25	81
9	96	100	88	85	67	54	49	14	80	95	98	88	85	68	58	52	23	80
10	96	100	87	84	66	51	49	9	79	94	98	87	84	67	55	52	19	79
11	96	100	87	81	66	51	49	6	78	94	98	87	82	67	55	52	16	79
12	94	100	87	81	63	51	49	6	77	93	98	87	82	65	55	52	16	78
13	94	100	87	76	63	51	49	6	76	93	98	87	77	65	55	52	16	77

**Table 3 jcm-11-05708-t003:** Cause-specific survival (CSS) function in percentages by years of follow-up and molecular classification based on the actuarial method, before and after multiple imputation (MI) (m = 5).

	Original Data Set N = 2235						Imputed Data Set N = 2867					
Year	Luminal A	Luminal B	Luminal with Ki67 Unknown	HER-2 Enriched	Triple Negative	Total	Luminal A	Luminal B	Luminal with Ki67 Unknown	HER-2 Enriched	Triple Negative	Total
1	99	97	98	95	92	97	98	97	97	91	92	96
2	97	95	95	88	86	94	95	94	95	85	87	93
3	97	92	93	84	83	92	95	91	92	81	84	91
4	96	89	91	82	79	89	93	88	90	79	80	88
5	95	87	89	80	77	87	93	86	88	77	78	87
6	94	84	86	76	75	85	92	84	86	74	76	84
7	94	82	84	75	73	83	91	82	83	73	74	82
8	93	79	83	73	73	82	90	79	82	71	74	81
9	93	78	81	68	72	80	90	79	81	67	73	80
10	93	75	80	68	72	79	90	76	80	67	73	79
11	93	75	80	68	72	79	90	76	80	67	73	79
12	93	75	78	68	72	78	90	76	78	67	73	78
13	93	69	78	68	72	77	90	71	78	67	73	77

**Table 4 jcm-11-05708-t004:** 5-year cause-specific survival (CSS) and relative survival (RS) by stage and molecular classification before and after multiple imputation (MI) (m = 5).

Variables	Original Data Set		Imputed Data Set	
	CSS	RS	CSS	RS
**Stage**	**N = 2387**		**N = 2867**	
IA	98	98.5	97	98.2
IB	100	99.8	98	99.1
IIA	94	95.8	93	95.7
IIB	93	94.6	92	94.0
IIIA	80	79.9	79	79.9
IIIB	71	77.7	73	81.6
IIIC	63	65.9	64	66.8
IV	31	31.1	39	39.6
**Molecular classification**	**N = 2235**		**N = 2867**	
Luminal A	95	96.4	93	94.6
Luminal B	87	88.4	86	88.3
Luminal with ki67 unknown	89	91.3	88	91.0
HER-2 enriched	80	81.5	77	78.6
Triple negative	77	75.9	78	77.9

**Table 5 jcm-11-05708-t005:** Competing-risks regression model of breast cancer before (Model 1) and after (Model 2) multiple imputation (MI) (m = 5).

	Model 1 (Original Data Set) N = 1924				Model 2 (Imputed Data Set) N = 2787			
Variables	Subhazard Ratio	St. Err.	*p*	CI 95%	Subhazard Ratio	St. Err.	*p*	CI 95%
**Age (ref. 55–64)**								
15–44	1.11	0.20	0.548	0.78, 1.59	1.21	0.20	0.240	0.88, 1.67
45–54	1.06	0.19	0.737	0.75, 1.51	1.15	0.18	0.374	0.84, 1.57
65–74	1.42	0.27	0.063	0.98, 2.07	1.47	0.25	0.022	1.06, 2.06
75 or more	2.78	0.48	<0.001	1.98, 3.89	2.72	0.43	<0.001	1.99, 3.71
**Sublocation (ref. Extern lower quadrant)**								
Nipple/Central region	1.04	0.30	0.894	0.58, 1.85	0.97	0.25	0.924	0.58, 1.63
Intern upper quadrant	1.96	0.57	0.020	1.11, 3.46	1.20	0.31	0.480	0.72, 2.00
Intern lower quadrant	0.94	0.37	0.878	0.44, 2.02	1.00	0.44	0.997	0.39, 2.56
Extern upper quadrant	1.34	0.33	0.244	0.82, 2.18	1.10	0.24	0.647	0.72, 1.68
Axillary	0.48	0.29	0.220	0.15, 1.56	0.63	0.64	0.660	0.06, 6.75
More than one location	1.60	0.39	0.051	1.00, 2.58	1.23	0.27	0.351	0.79, 1.90
**Stage (ref. IA)**								
IIA	2.27	0.56	0.001	1.40, 3.69	1.80	0.46	0.034	1.05, 3.07
IIB	3.25	0.82	<0.001	1.97, 5.34	2.46	0.64	0.002	1.44, 4.22
IIIA	8.65	2.05	<0.001	5.42, 13.78	5.53	1.47	<0.001	3.13, 9.79
IIIB	8.70	2.56	<0.001	4.89, 15.47	5.84	1.96	<0.001	2.85, 11.93
IIIC	12.40	3.49	<0.001	7.14, 21.53	7.77	2.34	<0.001	4.14, 14.60
IV	45.75	10.93	<0.001	28.64, 73.09	21.78	5.43	<0.001	12.83, 36.95
**Laterality (ref. Left)**								
Right	1.22	0.14	0.089	0.97, 1.54	1.10	0.12	0.356	0.89, 1.36
Bilateral	1.74	0.37	0.009	1.15, 2.64	1.38	0.39	0.278	0.75, 2.54
**Molecular classification (ref. Luminal with Ki67 unknown)**								
Luminal A	0.38	0.09	<0.001	0.24, 0.62	0.62	0.22	0.222	0.26, 1.44
Luminal B	1.01	0.15	0.950	0.76, 1.34	1.03	0.20	0.897	0.66, 1.58
Her-2 enriched	1.56	0.34	0.038	1.02, 2.38	1.31	0.29	0.239	0.83, 2.06
Triple negative	2.69	0.42	<0.001	1.98, 3.66	1.89	0.30	<0.001	1.38, 2.59

## Data Availability

Data available on request due to privacy/ethical restrictions.

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
