# Peer review of "Survival of Breast Cancer by Stage, Grade and Molecular Groups in Mallorca, Spain"

_jcm, 2022, doi:10.3390/jcm11195708_

Round 1

Reviewer 1 Report

In the manuscript authors assessed cause-specific survival  depending on the clinical and molecular data in  2869 breast cancer patients in the Mallorca area of Spain.  This  is a population-based retrospective follow-up study to which clinical data were obtained from the Mallorca Cancer Registry. Therefore, no clinical data are available for some patients.

To complete the missing data, the authors used the Multiple imputation (MI) method. The paper presents important data from this population. The results presented by the authors were previously published in literature in studies on other populations. The authors confirmed that patients with triple negative breast cancers, at a more advanced stage, and patients over 65 years old have worse prognosis in the Majorcan population. The following points should be considered:

1. Please clarify: In Table 1 HER2 positive patients account for 12% of the group (356), but below the HER-2 enriched molecular subtype is only 4.5% (130) . How is it possible?

2. In the case of molecular subtypes of breast cancer, there is no Luminal with Ki67 unknown subtype, which was identified in the publication and constitutes 34.8% of the studied group. Moreover, in 22% of patients it is not possible to define the molecular subtype due to lack of data, mainly Ki 67 status. Together, they constitute the majority of the studied group (56.8%), where the lack of data does not allow for the determination of the molecular subtype. In my opinion, such a large amount of missing data does not allow the assessment of survival depending on molecular subtypes. The actual subtype determination was possible in less than half of the cases. For this reason, an analysis of survival in molecular subtypes should not be performed. I recommend performing the analysis of survival depending on the molecular subtype only in the subgroup of patients with full medical data (ER; PR; HER2; Ki 67).

3. Is it possible to perform an additional analysis of cause-specific survival by age groups up to 49 years of age, from 50 to 69 years of age (screening programme) and over 70 years of age. Such an analysis may show whether the survival of a patient at the age undergoing screening programme is better.

Reviewer 2 Report

General

The paper presented does not show real new facts about prognostic and survival factors in breast cancer patients. This is caused by the fact, that even the best and sophisticated statistical methods cannot help when the main primary data situation is  weak:

In 69,5 percent of cases Ki67 values are missing, thus leading to the molecular group “luminal without Ki67”!

Furthermore the treatment modalities could not be reflected in all cases.

Both deficiencies lead to the result, that no real new data can be presented, and only known prognostic and survival factors are recapitulated. Therefore the presentation seems not to be useful and informative for an international readership and may be only of regional or national interest.

Specific

Page 4, table 1: correct lines.

Line 38: What is the difference between “age” and “age at diagnosis”?

Line 60: When the study is called “population based” than the number of DCO cases and cases without follow-up should be shown.

Round 2

Reviewer 2 Report

The authors implement  all recommondations of the reviewer very good